# An Electrochemical Impedance Spectroscopy-Based Aptasensor for the Determination of SARS-CoV-2-RBD Using a Carbon Nanofiber–Gold Nanocomposite Modified Screen-Printed Electrode

**DOI:** 10.3390/bios12030142

**Published:** 2022-02-25

**Authors:** Mahmoud Amouzadeh Tabrizi, Pablo Acedo

**Affiliations:** Electronic Technology Department, Universidad Carlos III de Madrid, 28911 Leganes, Spain; mamouzad@ing.uc3m.es

**Keywords:** electrochemical impedance spectroscopy, aptasensor, CNF–AuNP, SARS-CoV-2-RBD

## Abstract

Worldwide, human health is affected by severe acute respiratory syndrome coronavirus 2 (SARS-CoV-2). Hence, the fabrication of the biosensors to diagnose SARS-CoV-2 is critical. In this paper, we report an electrochemical impedance spectroscopy (EIS)-based aptasensor for the determination of the SARS-CoV-2 receptor-binding domain (SARS-CoV-2-RBD). For this purpose, the carbon nanofibers (CNFs) were first decorated with gold nanoparticles (AuNPs). Then, the surface of the carbon-based screen-printed electrode (CSPE) was modified with the CNF–AuNP nanocomposite (CSPE/CNF–AuNP). After that, the thiol-terminal aptamer probe was immobilized on the surface of the CSPE/CNF–AuNP. The surface coverage of the aptamer was calculated to be 52.8 pmol·cm^−2^. The CSPE/CNF–AuNP/Aptamer was then used for the measurement of SARS-CoV-2-RBD by using the EIS method. The obtained results indicate that the signal had a linear–logarithmic relationship in the range of 0.01–64 nM with a limit of detection of 7.0 pM. The proposed aptasensor had a good selectivity to SARS-CoV-2-RBD in the presence of human serum albumin; human immunoglobulins G, A, and M, hemagglutinin, and neuraminidase. The analytical performance of the aptasensor was studied in human saliva samples. The present study indicates a practical application of the CSPE/CNF-AuNP/Aptamer for the determination of SARS-CoV-2-RBD in human saliva samples with high sensitivity and accuracy.

## 1. Introduction

Coronavirus disease2019 (COVID-19) as an ongoing global infectious disease outbreak has caused a catastrophic impact on the world economy and healthcare systems. For this reason, there is a growing interest in the fabrication of a highly selective and sensitive device to diagnose the related virus that is called severe acute respiratory syndrome coronavirus 2 (SARS-CoV-2). By now, several methods have been reported for the sensitive detection of the SARS-CoV-2 in the literature, such as gene-based [1,2], immune-based [3,4] aptamer-based [5,6], and molecularly imprinted polymer-based assays [7,8]. The gene-based method is the most widely used [6,9,10]. In the gene-based method, the nucleotide chain of the SARS-CoV-2 must be first extracted and then detected. Although this method has very high sensitivity, it suffers from several drawbacks, such as being time-consuming, expensive (for a point-of-care device), and needing an operator with a high level of experience.

The immune-based method is performed in two different cases. In the first case, the ACE-2 antibody is immobilized on the surface of a transducer to detect the spike protein of the SARS-CoV-2 [4,11,12,13,14,15]. In the second case, the antigen is immobilized on the surface of a substrate (antigen sorbent assay) to detect antibody levels [16].

In the aptamer-based method, an aptamer is used to detect SARS-CoV-2-related proteins, such as the receptor-binding domain (RBD) [17,18], spike protein [19,20], and the nucleocapsid protein (Np) [21,22,23]. 

Aptamer-based assay methods have several advantages over immune-based and gene-based assays, such as being low cost, having high stability, being easy to modify, and, in some cases, it has better specificity and affinity [24].

In addition, electrochemical methods have several advantages, such as being low cost, having high sensitivity, their ease of use and operation, and having the potential to be integrated with point-of-care devices [25,26,27,28,29]. Among them, electrochemical impedance spectroscopy has a good potential to detect SARS-CoV-2 [30,31,32,33,34]. 

For these reasons, in this paper, we design an electrochemical impedance spectroscopy-based aptasensor to detect the SARS-CoV-2-RBD. To this end, a carbon nanofiber–gold nanoparticle (CNF–AuNP) nanocomposite is used as a modifier. The CNF–AuNP nanocomposite has several advantages, such as having high conductivity, high surface area, biocompatibility, high electrochemical and thermal stability [35], and being able to interact with thiol-terminal bio-recognizers, such as aptamers. To take advantage of this, the thiol-terminal aptamer is used for the fabrication of the aptasensor. After the modification of the carbon screen-printed electrode (CSPE) with CNFs-AuNPS, the thiol-terminal aptamer probes were immobilized on the surface of the CSPE/CNFs-AuNPs via an Au–S covalent bond. The obtained results show that the fabricated aptasensor has a high affinity and sensitivity to the SARS-CoV-2-RBD compared to previous aptasensors [17,18]. Additionally, the fabricated aptasensor showed high selectivity and stability. The results proved the practical application of the CSPE/CNFs-AuNPs in human saliva samples.

## 2. Materials and Methods

### 2.1. Reagents and Chemicals

All chemicals were of analytical reagent grade and used without further purification. Double deionized (DI) water (18.6 MΩ) was used throughout the research work. The thiol terminal aptamer was purchased from Nzytech (Lisboa, Portugal) with a sequence SH-(CH_2_)_6_−CAG CAC CGA CCT TGT GCT TTG GGA GTG CTG GTC CAA GGG CGT TAA TGG ACA-3′ [14]. Syndrome coronavirus-2 receptor-binding domain (SARS-CoV-2-RBD), human serum albumin (HSA), human immunoglobulin A (HIgA), human immunoglobulin M (HIgM), human immunoglobulin G (HIgG), hemagglutinin (HA), neuraminidase (N), sodium tetrachloroaurate (NaAuCl_4_), sodium borohydride (NaBH_4_)_,_ phosphoric acid (H_3_PO_4_), potassium hydroxide (KOH), potassium ferricyanide (K_3_Fe(CN)_6_), potassium ferrocyanide (K_4_Fe(CN)_6_), chitosan, and hexaammineruthenium (III) chloride ([Ru(NH_3_)_6_]Cl_3_) were obtained from Sigma-Aldrich (St. Louis, MO, USA). Dual working screen-printed electrodes (Ref X1110) were obtained from Metrohm-Drop Sens (Llanera, Spain).

### 2.2. Apparatus

The surface morphologies of the electrodes were characterized using scanning electron microscopy (SEM) (Field Electron and Ion (FEI)) (FEI, Hillsboro, OR, USA.). The cyclic voltammetry (CV) studies were performed using a potentiostat from Metrohm-DropSens Model µStat300 (Llanera, Spain). The elemental analysis was performed using an energy dispersive analysis of X-rays (EDX) (EDAX, Mahwah, NJ, USA). The attenuated total reflectance spectrum (ATR) study was performed by using a Nicolet iS50 Fourier transform infrared spectrometer (Thermo Fisher Scientific, Waltham, MA, USA). Electrochemical impedance spectroscopy (EIS) experiments were carried out using an ISX-3 impedance analyzer (Sciospec, Bennewitz, Germany) in phosphate-buffered saline (0.1 M PBS, pH 7.4) containing 5.0 mM Fe(CN)_6_^4−/3−^ couple (1:1) as the redox probe. An alternating current (AC) voltage of 10 mV and direct current (DC) voltage of 0.13 V were applied over a frequency range of 100 kHz to 0.1 Hz and the output signal was acquired using an EIS spectrum analyzer (EISSA) software.

### 2.3. Fabrication of the CSPE/CNF-AuNP/Aptamer

A total of 100 µL of NaAuCl_4_ (0.01 mM) were added into 9.8 mL of CNFs solution (0.025 g) and then mixed by ultrasonication for 60 min. After that, 100 µL of NaBH_4_ (0.01 mM) was added to the previous solution and mixed for another 60 min by ultrasonication. Then, CNFs–AuNPs were separated and dried. The fabricated CNF–AuNP nanocomposite was dispersed in a 10 mL chitosan (0.5%, acetate buffer pH 5.4) solution. The surface of the CSPE was then modified with 4 µL of the CNFs–AuNPs (12 µg, 0.1 M sodium acetate, pH 7.4) by the drop-casting technique and allowed to dry at room temperature. After that, the CSPE/CNFs–AuNPs was washed with DI water. Subsequently, 150 µL of thiol-terminal aptamer (120 nmol, 1 mM Mg^2+^, 0.1 M sodium acetate, pH 7.4) was dropped on the surface of the CSPE/CNFs–AuNPs for 24 h. During this time, the thiol-terminal aptamer probes interacted with AuNPs via the S–Au covalent bonds [36]. The CSPE/CNF–AuNP/Aptamer was then washed with 0.1 M PBS to remove loosely adsorbed aptamer. The CSPE/CNF–AuNP/Aptamer was immersed in 2% BSA solution (0.1M PBS, pH 7.4) for 60 min at room temperature to block nonspecific sites. The CSPE/CNF–AuNP/Aptamer was finally washed with DI water and stored in the refrigerator when not in use. Appendix A shows the schematic illustration of the CSPE/CNF–AuNP/Aptamer fabrication and sensing process.

### 2.4. Measurement Procedure of SARS-CoV-2-RBD in PBS

The typical EIS in the form of the Nyquist plot was used for the measurement of the SARS-CoV-2-RBD concentration in the measuring buffer (5 mM Fe(CN)_6_^3−/4−^, 0.1 M PBS, pH 7.4). 

During the measurement process, the CSPE/CNF–AuNP/Aptamer was first immersed in a PBS (0.1 M, pH 7.4) for 40 min at room temperature to record the blank signal. Then, the aptasensor was washed with a PBS (0.1 M, washing buffer) and then immersed in the measuring buffer to record the EIS signal. After that, the CSPE/CNF–AuNP/Aptamer was immersed in a PBS (0.1 M, pH 7.4) containing different concentrations of SARS-CoV-2-RBD for 40 min at room temperature. During this period, the SARS-CoV-2-RBD was incubated with aptamer probes. After that, the aptasensor was rinsed with washing buffer to wash away any loosely adsorbed SARS-CoV-2-RBD. The aptasensor was then immersed in the measuring buffer to record the EIS signal. In this way, as the SARS-CoV-2-RBD is a big molecule (35 kDa), the R_ct_ of the aptasensor will increase every time that aptasensor incubates with SARS-CoV-2-RBD. The standard deviation (SD) of the error bars for the ΔR_ct_ measurement were obtained by using four different aptasensors for a fixed concentration of SARS-CoV-2-RBD.

### 2.5. Data Processing and Statistical Analysis 

The semicircle diameter in the impedance spectrum equals the charge-transfer resistance (R_ct_). The R_ct_ is related to the charge transfer kinetics of the redox probe at the interface of the electrode. During the measurement process, the change in the R_ct_ of the aptasensor in the absence (blank) and the presence of a fixed concentration of SARS-CoV-2-RBD (ΔR_ct_) was recorded as the response of the CSPE/CNF–AuNP/Aptamer to the SARS-CoV-2-RBD in a solution. The EISSA software was used to analyze the Nyquist plots and obtain the ΔR_ct_ values for all the EIS measurements. To calculate the standard error bars, first, the SD of ΔR_ct_ from four different aptasensors was calculated. The calculated value was then divided by two (the square root of four measurements for the fixed concentration of SARS-CoV-2-RBD). So, the standard error bars were calculated with Equation (1):(1)$$\mathrm{Error}\;\mathrm{bar}=\mathrm{SD}\;\mathrm{of}\;\mathrm{the}\;\Delta R_{\mathrm{ct}}\;/\sqrt{n}\;(n=4)$$

Additionally, the relative standard deviation (RSD) was calculated by dividing the SD of ΔR_ct_ values by the mean values of the ΔR_ct_ Equation (2):RSD = SD of ΔR_ct_ /Mean of ΔR_ct_(2)

The limit of detection (LOD) was calculated by dividing the SD of the absolute value of R_ct_ related to the blank measurements (zero concentration of SARS-CoV-2-RBD) of four different aptasensors (σ) by the slope of the calibration curve (S). So, the detection limit was calculated with Equation (3):LOD = 3σ/S(3)

The value was then multiplied by three. All the statistical analysis were performed according to the literature [37].

### 2.6. Measurement Procedurein a Real Sample in Absence of SARS-CoV-2-RBD

First, the CSPE/CNF–AuNP/Aptamer was immersed in a PBS solution for 40 min and then washed. Subsequently, the EIS of the electrode was recorded in the measuring buffer (Appendix A, blue curve). The CSPE/CNF–AuNP/Aptamer was then immersed in 50 µL of 0.2 M PBS containing 50 µL of normal human saliva for 40 min at room temperature. After that, the aptasensor was washed with the washing buffer and then the EIS of the aptasensor was recorded in the measuring buffer. The PBS was added to the saliva sample to change its pH to 7.4 (the optimum pH) and increase its ionic conductivity (ionic conductance) for the electrochemical measurement. 

### 2.7. Measurement Procedurein Real Sample in Presence of SARS-CoV-2-RBD

In this process, two different concentrations of SARS-CoV-2-RBD were spiked in the saliva sample to measure them. To this end, 2 mL of normal human saliva sample was diluted with 1.8 mL PBS (0.1 M, pH 7.4). Then, the solution was mixed and divided into two vial samples. A total of 0.1 mL of 2.0 nMSARS-CoV-2-RBD solution was then added to the first vial and 0.1 mL of 16 nM SARS-CoV-2-RBD was added into the second vial. The samples were mixed for 30 min. After that, an aptasensor was immersed in the first vial (1.0 nMSARS-CoV-2-RBD) and another aptasensor was immersed in the second vial (8.0 nM SARS-CoV-2-RBD), each one for 40 min at room temperature. The aptasensors were then taken out and washed with 0.1 M PBS (pH 7.4). Finally, the aptasensors were dipped in the measuring buffer to record the EIS (Appendix A).

## 3. Results

### 3.1. The Surface Characterization of the Electrodes

The SEM images of a CSPE (A, B) and a CSPE/CNFs–AuNPs (C–F) are shown in Figure 1. As can be seen, the surface morphology of the CSPE before and after modification with CNFs–AuNPs changed. The presence of the AuNPs allows the thiol-terminal aptamers to be immobilized on the surface of the electrode. The average diameter of the CNFs and AuNPs was 101 nm and 10.1 nm, respectively.

Elemental analyses of CSPE/CNFs–AuNPs (a) and CSPE/CNF–AuNP/Aptamer (b) were also carried out by EDX technique (Figure 2A). The EDX spectrum of a CSPE/CNFs–AuNPs shows the presence of the gold element (Figure 2(Aa)). It indicates that the surface of the CSPE was properly modified with CNFs–AuNPs. After the immobilization of the thiol terminal aptamers, nitrogen, phosphor, and sulfur elements appeared in the EDX spectrum of the electrode (Figure 2(Ab)), indicating that the thiol-terminal aptamers were immobilized on the surface of the electrode.

The ATR spectrum of a CSPE/CNF–AuNP/Aptamer is also shown in Figure 2B. An absorption band at 3013 cm^−1^ due to the –NH_2_ stretching was observed. Additionally, an absorption band at 2622 cm^−1^ due to –CH stretching, an absorption band at 1609 cm^−1^ due to –C=C stretching, an absorption band at 1480 cm^−1^ due to in-plane vibration of nucleotide bases, an absorption band at 1333 cm^−1^ due to the –C–N stretching of nucleotide bases, an absorption band at 1221 cm^−1^ due to the –PO_2_^−^ stretching, an absorption band at 1163 cm^−1^ due to the –C–O stretching of aliphatic ether in deoxyribose, an absorption band at 965 cm^−1^ due to PO_2_^−^ bending, an absorption band at 890 cm^−1^ due to the deoxyribose ring vibration, and an absorption band at 710 cm^−1^ due to sugar-phosphate vibration are clearly seen [38]. The ATR results indicated that the aptamer probes were immobilized on the surface of the electrode.

### 3.2. Electrochemical Activity of the Modified Electrode

The electroactive surface area (A_eas_) and the roughness factor (RF) of the CSPE and the CSPE/CNF–AuNP were also obtained in a 5 mM Fe(CN)_6_^4−/3−^ solution (0.1 M PBS, pH 7.4) as the redox probe. The results show that the CNFs–AuNPs increase the electroactive surface area (A_eas_) of the electrode from 0.049 to 0.24 cm^−2^, the roughness factor (RF) of the electrode from 0.78 to 4.0, and the heterogeneous electron transfer constant (k_0_) of the probe from 0.004 cm·s^−1^ to 0.11 cm·s^−1^. It indicates that the CNFs–AuNPs nanocomposite significantly improved the electrochemical parameters of the electrode. More details are included in the supplementary section (Appendix A).

### 3.3. Study the Surface Coverage of the Immobilized Aptamer

The surface coverage of the aptamer (Γ_aptamer_)on the surface of the electrode was obtained according to the electrostatic interaction between Ru(NH_3_)_6_^3+^ as a positively charged redox probe and the negatively charged phosphate backbone of aptamer [39]. The surface coverage of the aptamer (Γ_aptamer_) was found to be 52.8 pmol·cm^−2^ (3.18 × 10^13^ molecules·cm^−2^). The relevant discussions and figures are given in the electronic supporting material (Appendix A).

### 3.4. Electrochemical Characterization of the CSPE/CNF–AuNP/Aptamer

The step-by-step electrochemical characterizations of the aptasensor were performed by EIS (Figure 3A) and CV (Figure 3B) in a 0.1 M PBS (pH 7.4) containing 5 mM Fe(CN)_6_^4−/3−^ as the redox probe. Figure 3A shows the EIS of the CSPE (a), the CSPE/CNFs–AuNPs (b), the CSPE/CNF–AuNP/Aptamer (c), and the CSPE/CNF–AuNP/Aptamer/SARS-CoV-2-RBD (d). As it can be seen in Figure 3A, after the modification of the CSPE (a) with CNFs–AuNPs (b), the diameter of the semicircular portion of the graph that is related to the R_ct_ decreased from 3048 Ω to 135.7 Ω, indicating that CNFs–AuNPs facilitated the charge transfer rate of the redox probe to the surface of the electrode. However, after the immobilization of the aptamer, the R_ct_ increased to 1186.6 Ω due to the electrostatic repulsion interaction between the negatively charged aptamer and the negatively charged electrochemical probe (Fe(CN)_6_^4−/3−^). This repulsion interaction hindered the electron transfer of the electrochemical probe to the surface of the electrode and, consequently, the R_ct_ increased. After the incubation of the SARS-CoV-2-RBD (64 nM) with the immobilized aptamer (d), the R_ct_ increased to 2432 Ω due to the mass-transfer limiting of Fe(CN)_6_^3−/4−^ to the electrode surface that is caused by a big molecule, such as SARS-CoV-2-RBD (~35 kDa) [13]. The details of the EIS spectrum are given in Appendix A.

Additionally, CVs (Figure 3B) studies showed that the intensity of the anodic and cathodic peaks increased after the modification of the CSPE (a) with CNFs–AuNPs (b). In addition, the difference between the potential of the anodic peak and the cathodic peak (ΔE=E_pa_ − E_pc_) decreased. As the aptamer probes were immobilized on the surface of the CSPE/CNFs–AuNPs (c), the intensity of these peaks decreased and the ΔE of them increased. After the incubation of the SARS-CoV-2-RBD with the immobilized aptamer probes (d), the intensity of the peaks and their ΔE decreased and increased more than before, respectively. All these changes in the EIS and CV characterizations of the electrode indicate that the electron transfer property of the CSPE was changed after each modification step.

### 3.5. Optimization of the Effective Parameters on the Response of the CSPE/CNF–AuNP/Aptamer 

The effects of the amount of the immobilized aptamer, the incubation time, and pH of the solution during the incubation on the response of the CSPE/CNF–AuNP/Aptamer to 5.0 nM of SARS-CoV-2-RBD (pH 7.4) were investigated in the measuring buffer (Appendix A). The following experimental conditions were found to give the best results: (a) 120 nmol of aptamer probe, (b) incubation time of 40 min, and (c) pH of 7.4 the solution. The relevant discussions and figures are provided in the supporting material.

### 3.6. Analytical Performance

Figure 4 shows the EIS (A) as well as the linear (B) and logarithmic corresponding calibration plot response (C) of the CSPE/CNF–AuNP/Aptamer to the different concentrations of SARS-CoV-2-RBD in the measuring buffer, respectively. As shown in Figure 4A, the R_ct_ of the biosensor increased as the concentration of SARS-CoV-2-RBD increased. The corresponding calibration plot shows that the response of the proposed biosensor had a logarithmic relationship with the concentration of SARS-CoV-2-RBD in the range of 0.01–64.0 nM (Figure 4B). The linear–logarithmic regression equation of the calibration curve (Figure 4C) is expressed as Equation (4):ΔRct (kΩ) = 0.313Log C[SARS-CoV-2-RBD] (nM) + 1.8(4)

The LOD of the CSPE/CNF–AuNP/Aptamer was found to be 7.0 pM.

According to the literature, there are between 25 and40 spike proteins on the surface of a SARS-CoV-2 molecule [40]. Since each spike protein of SARS-CoV-2 has an RBD, therefore, the LOD of the CSPE/CNFs-AuNPs/Aptamer would be 1.05–1.69 × 10^5^ copies/µL using Equation (5):(5)$$\mathrm{Copies}\;\mathrm{of}\;\mathrm{virus}/L=\frac{\mathrm{Molarity}\;\mathrm{of}\;\mathrm{SARS}-\mathrm{CoV}-2-\mathrm{RBD}\;}{\mathrm{Number}\;\mathrm{of}\;\mathrm{RBD}\;\mathrm{in}\;a\;\mathrm{Virus}}\times \mathrm{Avogadro}’s\;\mathrm{number}$$

The Langmuir-typical adsorption system (Equation (6)) [41,42] was used to calculate the Langmuir isotherm constant (K_L_), dissociation constant (K_d_), and the maximum number of binding sites (R_ctmax_) for the CSPE/CNF–AuNP/Aptamer (Figure 4D):(6)$$\frac{C\;\mathrm{SARS}-\mathrm{CoV}-2-\mathrm{RBD}}{\mathrm{Rct}}=\frac{1}{\mathrm{KL}\times \mathrm{Rct}\;\max}+\frac{C\;\mathrm{SARS}-\mathrm{CoV}-2-\mathrm{RBD}}{\mathrm{Rct}\;\max}$$
where R_et_ is the steady-state signal after the addition of a biomarker.

The values of 1/R_ct max_ and 1/K_L_ × R_ctmax_ can be obtained from the slope and the intercept point of Figure 4D, respectively. The values of R_ct max_, K_L_, and K_d_ (1/K_L_) were found to be 2450 µA, 2.35 nM^−1,^ and 0.42nM, respectively. The value of K_d_ is lower than the previously reported [17,18], indicating that the CSPE/CNF–AuNP/Aptamer has a high affinity to SARS-CoV-2-RBD. 

Additionally, the Gibbs free energy of desorption bind for the SARS-CoV-2-RBD-Aptamer was found to be −53.4 kJ·mol^−1^ by using Equation (7) [43]: ΔG = 2.03 × R × T × log (K_d_)(7)
where R is the universal gas constant of 8.31 J·K^−1^·mol^−1^ and T is the temperature in Kelvin.

The analytical performances of the proposed aptasensor were compared with the other reported aptasensors for measuring the biomarkers related to SARS-CoV-2 (Table 1). As can be seen, the analytical performances of the proposed aptasensor are better than the others in most cases. Although the LOD of the gold electrode/Aptamer that was combined with a secondary labeled aptamer for SARS-CoV-2 nucleocapsid protein (2019-nCoV-NP) is lower than the proposed aptasensor for the SARS-CoV-2-RBD [23], however, the process of the measurement of 2019-nCoV-NP was more complicated, expensive, and time-consuming. 

### 3.7. Stability, Reproducibility, and Selectivity of the CSPE/CNFs-AuNPs/Aptamer

The selectivity of the CSPE/CNF–AuNP/Aptamer to 2 nM SARS-CoV-2-RBD was studied in the absence and presence of HIgG, HIgA, HIgM, and HSA (Figure 5A). No sensible interference was observed for 10^2^-fold quantities of biomarkers in the determination of 2.0 nM SARS-CoV-2-RBD. The effect of the possible interference of HA and N from an influenza A virus was also studied (Figure 5B). As it can be seen, the R_ct_ of the CSPE/CNF–AuNP/Aptamer to 2 nM SARS-CoV-2-RBD in the presence of 50 nMHA and neuraminidase N did not change significantly (ΔR_ct_ = 16 Ω, or 0.8%). 

The stability of the CSPE/CNF–AuNP/Aptamer was also evaluated (Figure 5C). After 14 days, no obvious change was observed in the EIS of the aptasensor to 2.0 nM.

The reproducibility of the CSPE/CNF–AuNP/Aptamer was also evaluated for the measurement of 2.0 nM of SARS-CoV-2-RBD with six different aptasensors. The relative standard deviation (RSD) was calculated to be 3.6%.

### 3.8. Analytical Application of the Modified Electrode

The EIS of the CSPE/CNF–AuNP/Aptamer was recorded in a human saliva sample to study the possibility of the interference effect of the biomarkers that exist in the human saliva sample (Appendix A). As it can be seen, the R_ct_ of the electrode did not change, indicating the biomaterials in the saliva sample did not affect the electrochemical properties of the aptasensor. 

The CSPE/CNF–AuNP/Aptamer was then used for the measurement of SARS-CoV-2-RBD in human saliva samples to consider its applicability. Appendix A shows the EIS of the aptasensor to 1 nM and 8 nM of SARS-CoV-2-RBD in human saliva samples. The recovery of the analysis was obtained to be 95.3% and 96.2% for samples 1 and 2, respectively. The SARS-CoV-2 RBD concentrations in the normal human saliva samples were estimated using Figure 4C. The results indicate that the CSPE/CNF–AuNP/Aptamer has a good potential to detect the SARS-CoV-2-RBD in the human saliva samples accurately. The results were considered in comparison with a fluorescence-based assay of SARS-CoV-2-RBD [17].

## 4. Conclusions

In this paper, a CNF–AuNP nanocomposite was used as a modifier to design an electrochemical aptasensor to detect the SARS-CoV-2-RBD. The CNF–AuNP nanocomposite not only improved the electrochemical parameters of the CSPE but also assisted in the immobilization of the thiol-terminal aptamer probes via gold–thiol covalent bonds. The responses of the aptasensor to different concentrations of SARS-CoV-2-RBD were recorded using the EIS method in the Fe(CN)_6_^4−/3−^ solution as a redox probe. As the SARS-CoV-2-RBD interacted with the aptasensor, the R_ct_ of the aptasensor increased. Hence, the proposed method is a signal-on-based measurement method. The proposed aptasensor showed very good analytical performances in terms of selectivity, sensitivity, excellent repeatability, and satisfactory long-term storage stability. Notably, the affinity of the proposed aptasensor to SARS-CoV-2-RBD was better than the previously reported aptasensors. Lastly, the aptasensor was able to detect SARS-CoV-2-RBD in a human saliva sample. Although the prepared aptasensor paves the way for the design of a tool for the diagnosis of SARS-CoV-2-RBD in a saliva sample, however, it has some disadvantages. First, the sensitivity of the proposed aptasensor is less than that of immunosensors. Second, the use of commercially available electrodes already places the research at a disadvantage in terms of expanding novelty. However, the reproducibility of the signals recorded with the commercially available electrodes is better than that of the non-commercial ones.

## Figures and Tables

**Figure 1 biosensors-12-00142-f001:**
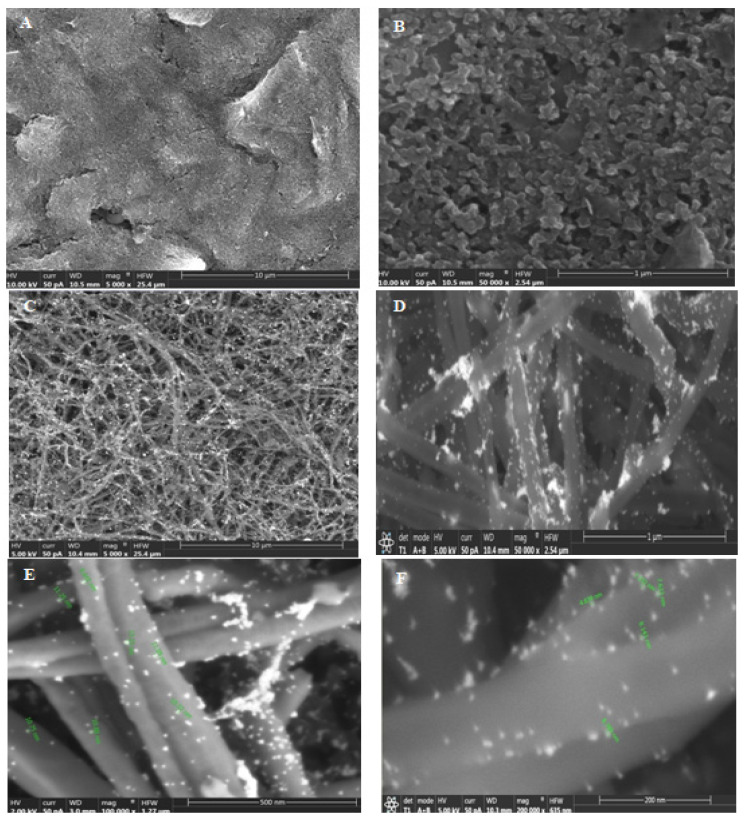
(**A**,**B**) SEM images of the CSPE and (**C**–**F**) CSPE/CNFs–AuNPs.

**Figure 2 biosensors-12-00142-f002:**
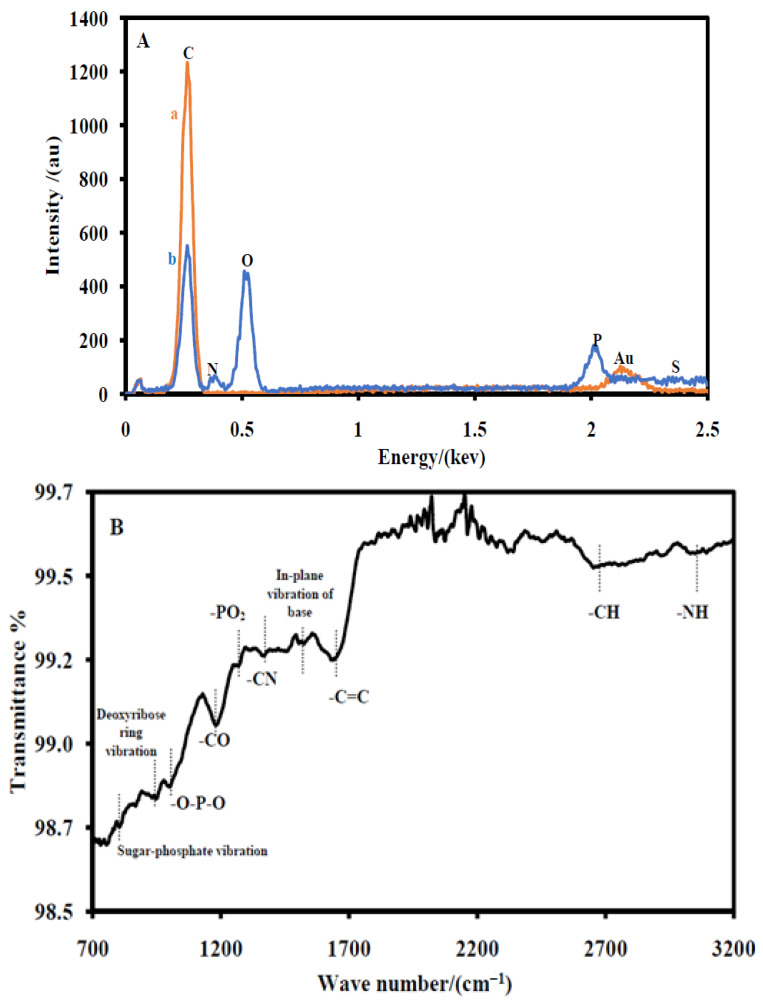
(**A**) EDS of the CSPE/CNFs–AuNPs (**a**) and the CSPE/CNF–AuNP/Aptamer (**b**). (**B**) ATR spectrum of the CSPE/CNF–AuNP/Aptamer.

**Figure 3 biosensors-12-00142-f003:**
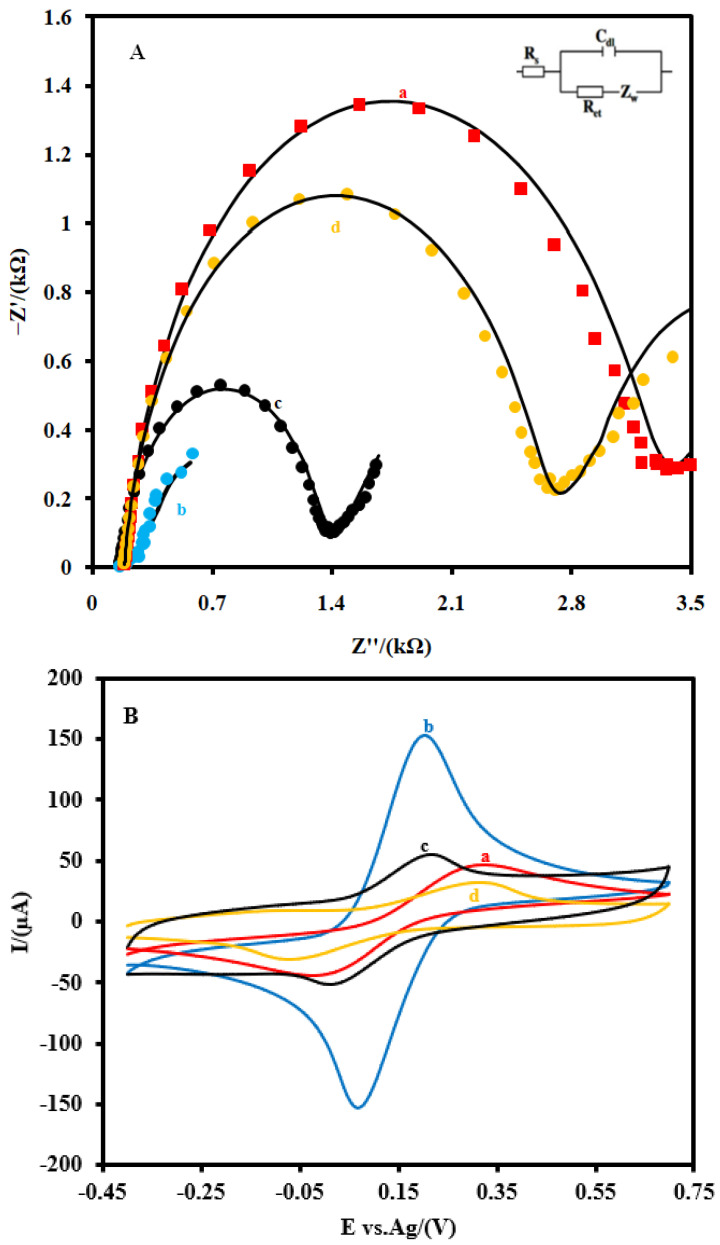
(**A**) EIS and (**B**) CVs and of the CSPE (**a**) the CSPE/CNFs–AuNPs (**b**), the CSPE/CNF–AuNP/Aptamer (**c**), and the CSPE/CNF–AuNP/Aptamer/SARS-CoV-2-RBD (**d**) at 0.1 M 5.0 mM Fe(CN)_6_^3−/4−^ solution (0.1 M PBS, pH 7.4). The equivalent electric circuit is compatible with the Nyquist diagrams. *R*_s_: Solution resistance; R_ct_: charge transfer resistance; C_dl_: Double layer capacitance; Z_w_: Warburg impedance. The AC amplitude voltage was 10 mV, DC voltage was 0.13 V, and frequency range was 100 kHz−0.1 Hz. The solid black lines indicate fitting curves. CVs were recorded at the scan rate of 0.05 V·s^−1^.

**Figure 4 biosensors-12-00142-f004:**
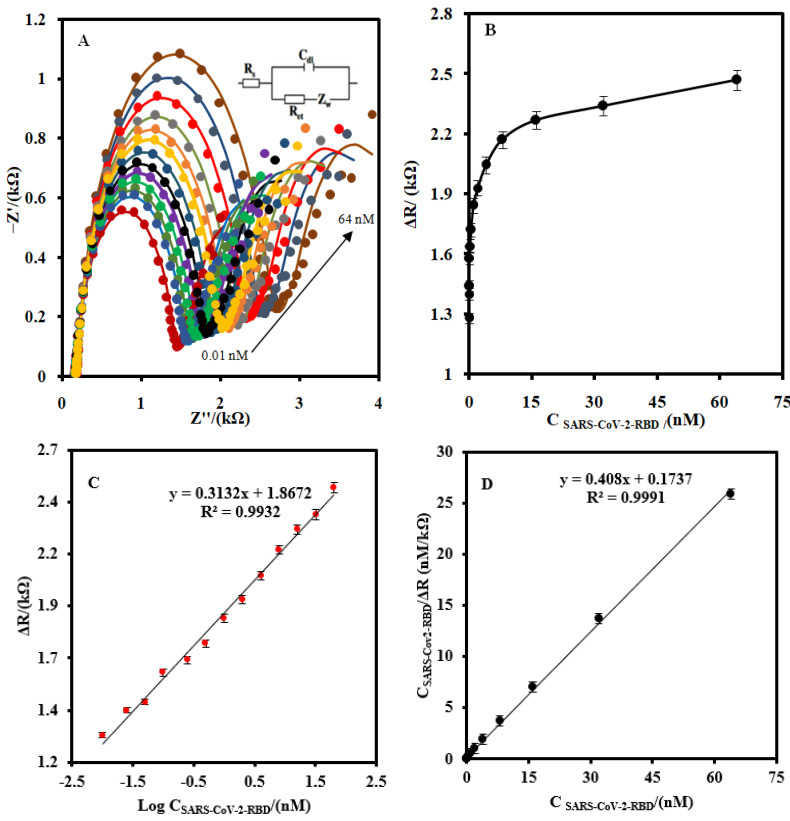
**(A**) The EIS of the CSPE/CNF–AuNP/Aptamer at the optimum operating conditions for different concentration of SARS-CoV-2-RBD (0.01, 0.025, 0.05, 0.1, 0.25, 0.5, 1.0, 2, 4, 8, 16, 32, and 64 nM) at 5.0 mM Fe(CN)_6_^3−/4−^ solution (0.1 M PBS, pH 7.4). (**B**) The linear-linear and (**C**) linear-logarithmic calibration curve plots. (**D**) The plot of the Langmuir binding isotherm model. The equivalent electric circuit is compatible with the Nyquist diagrams. R_s_: Solution resistance; R_ct_: charge transfer resistance; C_dl_: Double layer capacitance; Z_w_: Warburg impedance. The AC amplitude voltage was 10 mV, DC voltage was 0.13 V, and the frequency range was 100 kHz–0.1 Hz. The error bars were obtained by using four different aptasensors.

**Figure 5 biosensors-12-00142-f005:**
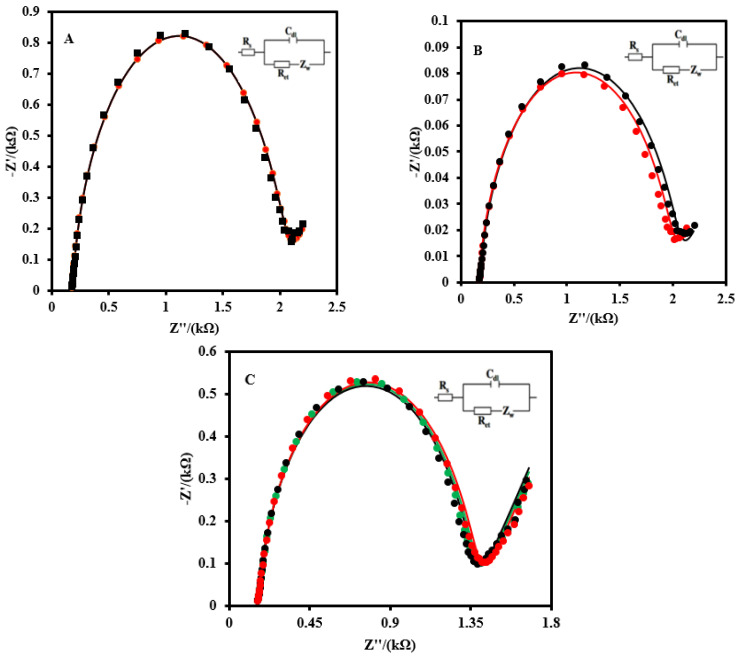
(**A**) EIS of the CSPE/CNF–AuNP/Aptamer to 2 nM SARS-CoV-2 RBD in the absence (black curve) and presence of 200 nM of HIgG, HIgA, HIgM, and HSA (red curve). (**B**) The EIS of the CSPE/CNF–AuNP/Aptamer to 2 nM of SARS-CoV-2 RBD in the absence (red curve) and presence of 200 nM of HA, and N from an influenza A virus (blue curve). (**C**) The stability of the CSPE/CNF–AuNP/Aptamer on the different days (First day: green curve; Seventh day: black curve; Fourteenth day: red curve). EIS spectrums were recorded in 5.0 mM Fe(CN)_6_^3−/4−^ solution (0.1M PBS, pH 7.4). The equivalent electric circuit is compatible with the Nyquist diagrams. R_s_: Solution resistance; R_ct_: Charge transfer resistance; C_dl_: Double layer capacitance; *Z*_w_: Warburg impedance. The AC amplitude was 10 mV, DC potential was 0.13 V, and the frequency range was 100 kHz−0.1 Hz.

**Table 1 biosensors-12-00142-t001:** Comparison of the analytical performance of the CSPE/CNF–AuNP/Aptamer with the other aptasensors and immunosensors for the biomarkers related to SARS-CoV-2.

Biosensor	Biomarker	Method	Linear Range	Limit of Detection	Response Time	Ref.
Carbon electrode/Graphene oxide/SP RBD	SP (78.3 kDa)	SWV	1–1000 ng/mL	0.11 ng/mL	45 min	[44]
CB/CSPE combined with Magnetic bead-based immunosensor	2019-nCoV-NP (51.1 kDa)	DPV	0.19–11.7 nM (0.01–0.6 μg/mL)	0.15 nM (8 ng/mL)	30 min	[11]
	SP (78.3 kDa)	DPV	0.04–10 μg/mL	19 ng/mL	30 min	
Gold electrode/Aptamer combined with a labeled aptamer	2019-nCoV-NP (51.1 kDa)	DPV	0.00048–0.97 nM (0.025–50 ng/mL) (0.0048–0.97 nM)	0.00016 nM (8.33 pg/mL)	60 min	[23]
Pad/Aptamer	2019-nCoV-NP (51.1 kDa)	SPR	0.5–16 ng/mL	1 ng/mL (20 pM)	110 min	[21]
ITO/gC_3_N_4_-CdS/Chitosan-Aptamer	SARS-CoV-2-RBD (35 kDa)	PEC	0.5–32 nM 17.5–1120 nM	0.12 nM 4.2 ng/mL	40 min	[18]
CSPE/Graphene/Spike IgG antibody			0.25 fg/mL–1 ng/mL	0.25 fg/mL	5	[12]
CSPE/CNF–AuNP/Aptamer	SARS-CoV-2-RBD (35 KDa)	EIS	0.01–64 nM (0.35–2240 ng/mL)	0.007 nM (0.24 ng/mL)	40 min	This work

SP: Spike protein; CB/CSPE: Carbon black-modified screen-printed electrode; ITO: Indium Tin Oxide electrode;g-C_3_N_4_*:* Graphitic carbon nitride; CdS: Cadmium sulfide; PEC: Photo electrochemical method; SPR: Surface plasmon resonance; SWV: Square-wave voltammetry; 2019-nCoV-NP: SARS-CoV-2 nucleocapsid protein; DPV: Differential pulse voltammetry.

## Data Availability

Appendix A associated with this article can be found in the online version that includes Appendix A.

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
