# Peer review of "An Electrochemical Impedance Spectroscopy-Based Aptasensor for the Determination of SARS-CoV-2-RBD Using a Carbon Nanofiber–Gold Nanocomposite Modified Screen-Printed Electrode"

_biosensors, 2022, doi:10.3390/bios12030142_

Round 1

Reviewer 1 Report

The ms reports the use of electrochemical impedance spectroscopy-based aptasensor for the determination of SARS-CoV-2-RBD. The proposed approach is based on manufacturing  carbon fiber modified gold nanoparticles, followed by surface immobilization of aptamer. Next the aptasensor was used to detect SARS-CoV-2RBD using electrochemical impedance spectroscopy. Although the results are convincing the structure of the paper and the selected data does not help the reader to appreciate the validity of the methods. several important results are in the supplementary files while it should be in the main text. I encourage the author's to reconsider the selected data to be presented within the manuscript file. In addition, further scientific discussion are necessary to be added especially for the electrochemical results.     

Author Response

Dear Referee,
Thank you very much for concerning our manuscript “ biosensors-1585339”.
We are highly thankful to you for having given us valuable suggestions for the improvement of our manuscript; they are helpful indeed. We resubmitted a list of the changes made to the manuscript, following your comments point by point (see next page). With this, we hope that this study now meets the criteria for publication in Biosensors.

Sincerely yours

Mahmoud Amouzadeh Tabrizi, PHD

Electronic Technology Department, Universidad Carlos III de Madrid, Madrid, Spain

Reviewer 1

The ms reports the use of electrochemical impedance spectroscopy-based aptasensor for the determination of SARS-CoV-2-RBD. The proposed approach is based on manufacturing  carbon fiber modified gold nanoparticles, followed by surface immobilization of aptamer. Next the aptasensor was used to detect SARS-CoV-2RBD using electrochemical impedance spectroscopy. Although the results are convincing the structure of the paper and the selected data does not help the reader to appreciate the validity of the methods. several important results are in the supplementary files while it should be in the main text. I encourage the author's to reconsider the selected data to be presented within the manuscript file.

Response: thank you for your advice. We moved the table S1 and Figure S5 to the manuscript

In addition, further scientific discussion are necessary to be added especially for the electrochemical results.

Response: Most of the discussion parts are in the supporting data such as roughness factor calculation, the surface coverage of the aptamer, electron transfer rate of the electrochemical probes, the optimization of the parameters and etc.

Thank you for taking your time

Reviewer 2 Report

The manuscript presents the development of an aptamer based sensor for the detection of the SARS-CoV-2-receptor binding domain through electrochemical impedance spectroscopy. The topic of the manuscript is within the Journal scope, ans may preset interest to the scientific community. The manuscript is in general well-written although language proof is necessary. But the originality and the novelty of the manuscript is doubtful since it presents a classical design, largely and easily encountered in the literature, in which the aptamer is immobilized through thiol chemistry on gold or gold modified electrodes and the binding event translated through the redox properties of hexacyanoferrates. This is however not the best practice (although largely encountered and used in many publications). Nonetheless, the introduction section appears to be prepared in hurry and lacks important citations to previous research on the development of electrochemical biosensors and assays for the detection of Sars-CoV2. Some other issues are:

- the determination of surface coverage was performed through “guanine base-methylene blue (G-MB) interaction”. This is not reliable and other methods for the determination of surface coverage must be used;

- impedance spectra must be fitted with an equivalent circuit and the fitting curves must be plotted together with the experimental data points; the quality of the fir must be provided/tabled together with the other determined parameters;

- experiments must be performed with other electroactive indicators.

If the authors are willing to consider these comments, I may reconsider my decision.

Author Response

Dear Referee,

Thank you very much for concerning our manuscript “ biosensors-1585339”.

We are highly thankful to you for having given us valuable suggestions for the improvement of our manuscript; they are helpful indeed. We resubmitted a list of the changes made to the manuscript, following your comments point by point (see next page). With this, we hope that this study now meets the criteria for publication in Biosensors.

Sincerely yours

Mahmoud Amouzadeh Tabrizi, PHD

Electronic Technology Department, Universidad Carlos III de Madrid, Madrid, Spain

Reviewer 2

The manuscript presents the development of an aptamer based sensor for the detection of the SARS-CoV-2-receptor binding domain through electrochemical impedance spectroscopy. The topic of the manuscript is within the Journal scope, ans may preset interest to the scientific community. The manuscript is in general well-written although language proof is necessary. But the originality and the novelty of the manuscript is doubtful since it presents a classical design, largely and easily encountered in the literature, in which the aptamer is immobilized through thiol chemistry on gold or gold modified electrodes and the binding event translated through the redox properties of hexacyanoferrates. This is however not the best practice (although largely encountered and used in many publications). Nonetheless, the introduction section appears to be prepared in hurry and lacks important citations to previous research on the development of electrochemical biosensors and assays for the detection of Sars-CoV2. Some other issues are:

- the determination of surface coverage was performed through “guanine base-methylene blue (G-MB) interaction”. This is not reliable and other methods for the determination of surface coverage must be used;

Response: I corrected it based on the interaction of ruthenium hexamine with aptamer. Kindly see Figure S3.

In addition, we added some papers about the SARS-CoV measurement that were published recently. Kindly see refs (2-6) and 19-20.

- impedance spectra must be fitted with an equivalent circuit and the fitting curves must be plotted together with the experimental data points; the quality of the fir must be provided/tabled together with the other determined parameters;

Response: we applied the fitting curves to all the EIS graphs (solid lines). We only put the data of Figure 3A in table S1 and add the fitting curves to all the figures but it is impossible to add a table for each figure that has EIS curves. I have never seen a paper that all the EIS graphs included a table. Table S1 and the screenshot of the software analysis are below.

Electrode

R1

R2

W1

P1

n

CSPE

192.2

3048.2

976.81

5.28e-7

0.918

CSPE/CNFs-AuNPs

93.2

135.79

248.7

1.6e-4

0.368

CSPE/CNFs-AuNPs/Aptamer

165.12

1186.6

450.2

6e-7

0.9

CSPE/CNFs-AuNPs/Aptamer/SARS-CoV-2-RBD

188.8

2432.8

751.9

5.4e-7

0.919

CSPE

CSPE/CNFs-AuNPs

CSPE/CNFs-AuNPs/Aptamer

CSPE/CNFs-AuNPs/Aptamer/SARS-CoV-2-RBD

- experiments must be performed with other electroactive indicators.

Response: I did not understand what you exactly wanted me to do. If you wanted me to use ruthenium hexamine chloride to calculate the surface coverage of aptamer? I did it before. But if you wanted me to replace the Fe(CN)63−/4− with another electrochemical probe with all due respect I cannot do that. I cannot do all the optimization and measurement processes again Fe(CN)63−/4−  is a comment probe for the measurement of a target via a label-free based sensor. If the authors are willing to consider these comments, I may reconsider my decision.

Thank you for taking your time

Reviewer 3 Report

The work "An electrochemical impedance spectroscopy-based aptasensor for the determination of SARS-CoV-2-RBD using carbon nanofiber-gold nanocomposite modified screen-printed electrode" is very well made and presented. It is very nice to see that the authors used aptamer as the selection and sensitive layer for the SARS-CoV-2-RBD recognition. 

Comments:

1. "2.7. Measurement procedure in real simple in presence of SARS-CoV-2-RBD"- Please change simple for sample

2. Line 357, please change florescence-based for fluorescence-based. 

3. Add to the Materials and Methods how did you store the ready-to-use electrodes. I found information that Au-S bound is not stable especially on air. 

4. Also it is known that ssDNA could physically adsorb on the gold surface. Did author carry out the experiment with aptamer for RBD without SH modification? What the results could be there?

Author Response

Reviewer 3

Comments:

  1. "2.7. Measurement procedure in real simple in presence of SARS-CoV-2-RBD"- Please change simple for sample

Response: it was corrected. Kindly see page 5, page 177.

  1. Line 357, please change florescence-based for fluorescence-based. 

Response: it was corrected. Kindly see page 15, page 383

  1. Add to the Materials and Methods how did you store the ready-to-use electrodes. I found information that Au-S bound is not stable especially on air. 

Response: we have mentioned on page 4, line 120. “The CSPE/CNFs-AuNPs/Aptamer was finally washed with DI water and stored in the refrigerator when not in use.”

  1. Also it is known that ssDNA could physically adsorb on the gold surface. Did author carry out the experiment with aptamer for RBD without SH modification? What the results could be there?

Response: the chemical attachment of the aptamer on the surface of the electrode via the S-Au covalence band is much stronger than the physical attachment of the aptamer on the surface of the electrode. The recognizer should not to released on the surface of the electrode. they are several parameters that will affect the physically immobilized aptamer probe such as electrolyte and rate of the flow rate of solution (if a flow cell setup is used to transfer the sample to the electrochemical cell to interact the target with biosensor). In addition, the affinity of the aptamer to the SARS-Cov-2-RBD is high and it is possible the SARS-Cov-2-RBD interacts with the aptamer and peels the aptamer off on the surface of the electrode, decreasing the resistance of the biosensors. it is also possible the SARS-Cov-2-RBD is attached to the physically adsorbed aptamer probe, sticking on the surface of the electrode, and then the resistance of the biosensor increase. Then we cannot explain what is going on the surface of the electrode. for one measurement, the signal will decrease but for another measurement, the signal increased. So it is better we immobilize the aptamer probe via a chemical bond and as you know the S-Au bond is one of the strongest bonds.      

Thank you for taking your time

Round 2

Reviewer 2 Report

Some of my questions were answered but a concern still remains. Apart from using a new aptamer for detection of SARS-CoV-2-RBD, the authors must explain which is the novelty of the manuscript, at least in terms of sensor's architecture.

Author Response

Dear reviewer

Thank you for taking the time to review our manuscript. the novelty of the manuscript has been mentioned at the end of the introduction section (61-70). we said "The CNF-AuNPs nanocomposite has several advantages such as high conductivity, high surface area, biocompatibility, high electrochemical and thermal stability [35], and can interact with thiol-terminal bio-recognizer such as aptamer. To take advantage of this, the thiol-terminal aptamer was used for the fabrication of the aptasensor. After the modification of the carbon screen-printed electrode (CSPE) with CNFs-AuNPS, the thiol-terminal aptamer probes were immobilized on the surface of the CSPE/CNFs-AuNPs via an Au-S covalent bond. The obtained results showed that the fabricated aptasensor had a high affinity and sensitivity to the SARS-CoV-2-RBD compared to the previous aptasensors [17,18]. Also, the fabricated aptasensor showed high selectivity and stability."

To us, that is the novelty of our manuscript. if you believe that is not the novelty, since you read the manuscript, please tell us based on the result and sensor's architecture that in it we use CNF-Au, what is the novelty. we will write it in the manuscript. to us, what was mentioned in lines 61-70 indicated the novelty.

Thank you again for your time 

Mahmoud Amouzadeh Tabrizi